# Sliding-Window CNN + Channel-Time Attention Transformer Network Trained with Inertial Measurement Units and Surface Electromyography Data for the Prediction of Muscle Activation and Motion Dynamics Leveraging IMU-Only Wearables for Home-Based Shoulder Rehabilitation

**DOI:** 10.3390/s25041275

**Published:** 2025-02-19

**Authors:** Aoyang Bai, Hongyun Song, Yan Wu, Shurong Dong, Gang Feng, Hao Jin

**Affiliations:** 1College of Information Science & Electronic Engineering, Zhejiang University, Hangzhou 310027, China; aoyang@zju.edu.cn (A.B.); dongshurong@zju.edu.cn (S.D.); 22nd Affiliated Hospital, School of Medicine, Zhejiang University, Hangzhou 310009, China; hedy.song@zju.edu.cn (H.S.); yanwu@zju.edu.cn (Y.W.); gangfeng@zju.edu.cn (G.F.)

**Keywords:** inertial measurement unit, surface electromyography, shoulder rehabilitation, deep learning, musculoskeletal analysis

## Abstract

Inertial Measurement Units (IMUs) are widely utilized in shoulder rehabilitation due to their portability and cost-effectiveness, but their reliance on spatial motion data restricts their use in comprehensive musculoskeletal analyses. To overcome this limitation, we propose SWCTNet (Sliding Window CNN + Channel-Time Attention Transformer Network), an advanced neural network specifically tailored for multichannel temporal tasks. SWCTNet integrates IMU and surface electromyography (sEMG) data through sliding window convolution and channel-time attention mechanisms, enabling the efficient extraction of temporal features. This model enables the prediction of muscle activation patterns and kinematics using exclusively IMU data. The experimental results demonstrate that the SWCTNet model achieves recognition accuracies ranging from 87.93% to 91.03% on public temporal datasets and an impressive 98% on self-collected datasets. Additionally, SWCTNet exhibits remarkable precision and stability in generative tasks: the normalized DTW distance was 0.12 for the normal group and 0.25 for the patient group when using the self-collected dataset. This study positions SWCTNet as an advanced tool for extracting musculoskeletal features from IMU data, paving the way for innovative applications in real-time monitoring and personalized rehabilitation at home. This approach demonstrates significant potential for long-term musculoskeletal function monitoring in non-clinical or home settings, advancing the capabilities of IMU-based wearable devices.

## 1. Introduction

The shoulder joint (SJ) is one of the most complex and functionally versatile joints in the human body, enabling a wide range of upper-limb movements [1,2]. Its structure, comprising the scapula, clavicle, and humerus, is supported by the rotator cuff and scapular stabilizer muscles, ligaments, and joint capsules that maintain stability and mobility [3,4]. The shoulder’s range of motion (ROM) extends from 120° to 180°, surpassing that of all other joints, and it plays a vital role in essential tasks like lifting, grasping, and fine motor activities [3,5]. Over 65% of the kinetic energy required for upper-limb function comes from the interaction between the SJ and scapula [6,7].

This highly mobile joint is subject to complex biomechanical loads, especially during dynamic movements. The rotator cuff and scapular stabilizer muscles cooperate to centralize and stabilize the SJ, but imbalances in muscle activity can result in dysfunction. Subacromial impingement syndrome (SAIS), a common cause of restricted mobility, often arises from insufficient coordination among these muscles [8,9]. The prevalence of rotator cuff injuries increases with age, affecting over 20% of individuals over 40, and over 50% of those over 65, hindering functional recovery [10]. The dysfunction of scapular stabilizers, such as in scapular dyskinesis (SD), can exacerbate shoulder instability, pain, and muscle weakness, significantly affecting daily life and work capacity [11,12].

Shoulder dislocation is another prevalent condition, with an incidence rate of 23 cases per 100,000 individuals and a recurrence rate of as high as 90% in younger populations [13,14]. Shoulder injuries are among the most common musculoskeletal disorders, with a prevalence of 55% and an annual incidence rate of 62 new cases per 1000 individuals [1,10]. These injuries not only affect a patient’s quality of life but also restrict their ability to perform daily, occupational, and recreational activities, imposing significant psychological and economic burdens [7,15,16].

Recent research on shoulder injuries has increasingly focused on understanding the biomechanical mechanisms that regulate shoulder stability and functional recovery. Studies have also supported the development of personalized rehabilitation strategies, including physical therapy, functional exercises, and biofeedback [17,18]. With increasing use of wearable devices such as electromyography (EMG) and inertial measurement units (IMU), the real-time monitoring of muscle activity and joint motion has become feasible [19,20]. When integrated with machine learning models, these devices can predict muscle activation patterns and joint motion trajectories, leading to more tailored and efficient rehabilitation strategies [21,22].

Several studies have demonstrated the benefits of combining EMG and IMU data for shoulder rehabilitation. Hasan et al. [22] introduced a neural network framework that integrates both data types to improve rehabilitation robotics, achieving robust predictions of upper-limb dynamics. Similarly, Lobo et al. [23] reviewed wearable technologies and highlighted the importance of deep learning for multisensor data fusion in motion analysis. Hua et al. [24] evaluated the use of machine learning models to classify upper limb movements using IMU data. While these models showeded high accuracy in controlled environments, their performance in real-world scenarios was challenged by the variability of patient movements. Brennan et al. [25] developed a cost-effective method for shoulder rehabilitation motion segmentation using a single IMU, but sensor configuration limitations led to suboptimal tracking accuracy for complex shoulder movements.

These advancements highlight the growing potential of machine learning in healthcare, particularly in intelligent rehabilitation systems. However, balancing model robustness, real-time performance, and cost-effectiveness remains a significant challenge. Deep neural networks (DNNs) and convolutional neural networks (CNNs) have been widely used for processing temporal signals, including EMG and IMU data for activity recognition and joint angle prediction, showcasing their considerable potential. Mekruksavanich et al. [9] proposed a residual deep learning model that integrates sensor data for fitness activity classification with exceptional accuracy, demonstrating deep learning’s potential in dynamic motion classification for shoulder rehabilitation. Werner et al. [21] used a linear neural network to estimate shoulder loading during wheelchair activities, further illustrating the versatility of deep learning across various rehabilitation contexts.

Shen et al. [26] introduced a multimodal data fusion approach that combines IMU and EMG signals with a long short-term memory (LSTM) model to assess upper limb functionality. While this method improves analytical precision by integrating multisource data, its reliance on high-quality sensor data and computationally intensive processes limits its real-time applicability. Ren et al. [27] combined IMU and EMG data using deep learning to classify fundamental shoulder movements during rehabilitation, although this method’s robustness decreased for complex multi-step rehabilitation tasks. Elkholy et al. [28] used DNNs to classify basic upper-limb activities, but the model’s limited adaptability to unstructured data restricts its broader clinical use.

The integration of transfer learning and hybrid models has accelerated progress in this field. Choi et al. [18] introduced a transfer learning-based system for detecting upper-limb movement intentions using EMG and IMU data, offering an adaptive solution to individual variability. This method illustrates the potential of transfer learning in addressing data distribution discrepancies. Wang et al. [29] developed a hybrid multi-feature neural network that integrates traditional feature extraction with deep learning for upper-limb gesture classification. While it enhanced model robustness, its high computational demand limits its application in resource-limited environments. Teng et al. [30] explored the use of graph convolutional networks (GCNs) to predict shoulder and elbow joint angles using EMG and IMU data. Although the model captured complex motion patterns effectively, its reliance on large-scale, high-quality training data presents challenges for real-world use.

Despite progress in shoulder rehabilitation technologies, developing robust models for real-time prediction remains a significant challenge. Issues include model adaptability to diverse patient data, conflicts between real-time and lightweight solutions, and concerns about multimodal data fusion robustness. Wei and Wu [20] discussed the integration of wearable sensors and machine learning for rehabilitation, emphasizing the importance of reliable systems. Foroutannia et al. [31] highlighted the need for lightweight, efficient models that maintain high accuracy across various patient populations to meet clinical demands.

This study introduces a novel neural network architecture for predicting EMG features from IMU data in shoulder rehabilitation systems. By incorporating advanced deep learning techniques and multimodal sensor fusion, the proposed system refines neural network architectures and data processing methods to achieve high-precision muscle activity prediction and real-time shoulder motion decoding. Compared to existing approaches, the model reduces computational overhead while improving real-time performance and adaptability, making it suitable for unstructured scenarios, such as home rehabilitation.

This research not only provides an efficient technical solution but also bridges the gap between innovation and clinical application. By enhancing the functionality of wearable devices, it offers clinicians more reliable assessment tools and lays the foundation for personalized rehabilitation strategies for shoulder injury patients. Ultimately, the findings aim to improve shoulder rehabilitation efficiency and promote the adoption of intelligent rehabilitation technologies.

## 2. Materials and Methods

### 2.1. Data Collection

To collect IMU and sEMG data from the SJ, this study developed an integrated experimental system designed to facilitate both the acquisition and processing of data. The system comprises two primary components: an sEMG data acquisition device and an IMU data acquisition device. The complete process of the acquisition system is illustrated in Figure 1.

The IMU acquistion device captures the kinematic and dynamic characteristics of the SJ. The single sensor device used is the WT9011DCL, manufactured by WitMotion Shenzhen Co. (Floor 3, Building 7, YunLi Park, Guangming District, Shenzhen, China). Its key components include a tri-axial accelerometer, a tri-axial gyroscope, and a tri-axial magnetometer, accompanied by modules for the acquisition, storage, and transmission of data. The accelerometer measures linear acceleration in three-dimensional space, the gyroscope records angular velocity changes, and the magnetometer detects magnetic field information to aid in posture recognition. The data sampling frequency is configured at 200 Hz, ensuring adequate temporal resolution to accurately capture SJ motion patterns. The collected data are wirelessly transmitted via a Bluetooth module to a computer, where they are synchronized with electromyography signals, facilitating the integrated analysis of dynamic motion characteristics and muscle activation patterns.

A custom EMG acquisition module was built, which is mainly composed of an AD1299 chip, which is a four-channel, 24 bit, Analog-to-Digital Converter for EEG and biopotential measurements. The device’s key components include a four-channel sEMG signal acquisition unit, signal amplifiers, filters, an analog-to-digital (AD) converter, and modules for data storage and transmission. High-sensitivity electrodes accurately detect the electromyographic activity of shoulder-related muscle groups. Due to the low amplitude of sEMG signals (typically ranging from a few microvolts to millivolts), the system is equipped with high-gain amplifiers to boost signal strength. Signal preprocessing involves high-pass filtering (cut-off frequency: 20 Hz) and low-pass filtering (cut-off frequency: 500 Hz) to reduce environmental noise and mitigate power-line interference. The filtered analog signals are digitized at a sampling frequency of 2000 Hz using the AD converter, delivering high-temporal-resolution inputs for subsequent analysis. The digitized signals are transmitted to a computer in real time through a serial port, ensuring reliable and complete data transmission.

To ensure temporal synchronization between the sEMG and inertial data, the system timestamps the acquired signals and conducts preliminary cleaning and calibration using dedicated software after acquisition. Figure 1c shows the row IMU and sEMG signal. These steps enable the system to deliver high-quality input data, providing a robust foundation for subsequent modeling and analysis.

The experiment recruited 25 volunteers, comprising 10 healthy participants and 15 individuals with SJ movement injuries, aged 18 to 30 years (mean age: 25.3 ± 3.1 years). Among the healthy participants, 7 were male and 3 were female, while the injury group consisted of 7 males and 8 females. The injuries included subacromial impingement syndrome, mild rotator cuff tears, and shoulder complex injuries related to fractures. Participant demographic information is summarized in Table 1. All participants had stable symptoms and successfully completed the prescribed experimental movements. All participants were right-handed and had no history of neurological disorders or acute musculoskeletal injuries. The experimental design and procedures were approved by the ethics committee, and all participants gave written informed consent.

The experiments were conducted in a standardized indoor environment with temperatures maintained between 22 °C and 25 °C to minimize environmental noise and interference from large equipment, ensuring stable experimental conditions. Participants were instructed to avoid intense physical activity or strenuous training before the experiment to maintain natural muscle relaxation and prevent fatigue from compromising data quality. For participants with shoulder movement injuries, the research team tailored the intensity and guidance of experimental movements to individual conditions, ensuring safety and comfort throughout the experiment. Before data collection, the target muscle areas were cleaned and hydrogel electrodes were placed at designated locations to minimize skin impedance. EMG signals were collected from key shoulder muscles, including the Deltoid (Del), Trapezius (Trap), Biceps (Bicep), Triceps (Tricep), and Latissimus dorsi (Lat). Inertial sensors were securely fastened near the SJ using straps to record three-dimensional motion data. The specific collection check point is shown in the Figure 1b.

The experimental procedure comprised multiple steps to address both static and dynamic data collection requirements. Initially, participants underwent baseline testing in a static posture to record EMG and inertial data from the SJ without movement, providing a reference for noise correction. Dynamic testing included three standardized SJ movements: abduction, forward flexion, internal and external rotation motion, as shown in Figure 2. Under researcher guidance, participants performed the movements at a fixed pace, repeating each movement 10 times, with each repetition lasting 5 s and a 2 s interval between repetitions. For specific movements, lightweight dumbbells were introduced as external loads to investigate muscle activation patterns under varying load conditions. For participants with shoulder injuries, the range of motion and speed were adjusted to ensure safety and minimize pain or discomfort.

The entire experiment lasted approximately 60 min, during which each participant’s movements were carefully guided and calibrated. After the experiment, all collected data were subjected to quality checks using specialized software, including signal cleaning, noise correction, and time alignment, to ensure compliance with the requirements for subsequent analysis. This high-quality dataset serves as a reliable foundation for investigating SJ movement characteristics and muscle activation patterns.

### 2.2. Data Process

Data processing in this study involves both EMG signals and IMU signals. Due to the significant differences in sampling frequencies between the two types of signals (EMG signals are sampled at a much higher rate than IMU signals), the EMG signals were downsampled to temporally align with the IMU data. To meet the structured input requirements of the model, the processed data were further segmented into fixed-length sliding windows, ensuring a consistent data format for subsequent analysis and modeling. The outcomes of the data processing are depicted in Figure 2.

#### 2.2.1. IMU Signal Processing

IMUs comprise a core structure consisting of a three-axis accelerometer and a three-axis gyroscope. These components measure three-axis acceleration and the angular velocity of joints during spatial motion, enabling the real-time analysis of joint posture angle variations in three-dimensional space. To extract motion characteristics, spatial angle transformations and feature extraction techniques were applied to the IMU signals.

The raw data recorded by the IMU sensors are represented in the local coordinate system of the sensor. For unified analysis, the local coordinate data must be transformed into the global coordinate system to enable the integration and comparison of data from different sensors. This transformation is performed using a rotation matrix R, defined as Equation (Equation 1):(1)$$g_{\mathrm{global}}=R\cdot g_{\mathrm{local}}$$

Here, $g_{\mathrm{global}}$ represents the global coordinate system, $g_{\mathrm{local}}$ represents the local coordinate system, and R is derived from quaternions, calculated by integrating gyroscope and accelerometer data to estimate orientation. The quaternion is computed using sensor fusion algorithms, such as the extended Kalman filter, to ensure accuracy in the transformation to the global coordinate system.

By integrating accelerometer and gyroscope data, the orientation angles of the SJ in three-dimensional space—roll (ϕ), pitch (θ), and yaw (ψ)—can be determined as follows:(2)$$\left\{\begin{array}{c} \phi =\arctan\left\{\frac{a_{y}}{a_{z}}\right\},\\ \theta =\arcsin\left\{-\frac{a_{x}}{g}\right\},\\ \psi =\arctan\left\{\frac{m_{y}}{m_{x}}\right\}, \end{array}\right.$$
where $a_{x},a_{y},a_{z}$ are accelerometer data, $m_{x},m_{y}$ are magnetometer data, and *g* represents gravitational acceleration.

#### 2.2.2. sEMG Signal Processing

Surface-EMG is a low-amplitude bioelectric signal, with its magnitude typically proportional to muscle force and generally ranging from 0 to 1.5 mV. Its analyzable frequency range extends from 0 to 500 Hz, with the majority of energy concentrated in the 20–150 Hzband. During acquisition, sEMG signals are vulnerable to noise interference, including baseline drift noise (1–4 μV) and industrial frequency interference (0–60 Hz). Therefore, filtering and denoising are crucial for extracting target features and enhancing signal quality.

The primary goal of preprocessing is to minimize industrial frequency noise and other interferences during acquisition, enhance the signal-to-noise ratio (SNR), and retain as much useful information as possible. Filtering is crucial in this process; this involves the design of specific frequency ranges that permit target signals to pass while attenuating unwanted frequencies. Among the software-based filtering techniques, the Butterworth filter is widely used. Renowned for its maximally flat amplitude response, this filter preserves signal integrity while avoiding the amplitude attenuation caused by filtering. Compared to first-order low-pass filters, the Butterworth filter delivers a superior performance and is available in various forms, including high-pass, low-pass, band-pass, and band-stop filters. It demonstrates stable amplitude-frequency characteristics both within and outside the passband, making it highly suitable for sEMG signal filtering. The transfer function of the Butterworth filter is presented in Equation (Equation 3):(3)$${\left|H(j\omega )\right|}^{2}=\frac{1}{1+{\left(\frac{\omega }{\omega_{c}}\right)}^{2n}}$$
where *n* represents the filter order, *j* is the imaginary unit, ω is the complex variable in the Laplace transform, and $\omega_{c}$ denotes the cutoff frequency. This paper adopts a fourth-order Butterworth filter for the band-pass filtering, and the passband boundary is 10–500 Hz.

### 2.3. Dataset Preparation

#### 2.3.1. Data Alignment

To construct a dataset for model training and testing, the collected high-frequency sEMG and IMU signals underwent downsampling, temporal alignment, sliding window segmentation, and feature extraction. The final dataset was split into training, validation, and test sets. This process ensured temporal synchronization, feature consistency, and balanced data splits, establishing a robust foundation for the generalization of subsequent models.

Because the EMG signal sampling frequency is 2000 Hz, which is significantly higher than the IMU signal sampling frequency of 200 Hz, downsampling of the EMG signals was required to achieve temporal alignment between the two signals. Downsampling the EMG signal inevitably results in the loss of some signal details, but it remains a reasonable and necessary operation. Key features of the EMG signal are extracted prior to downsampling, ensuring that essential information is preserved during the training process. Downsampling substantially reduces the time and computational cost associated with model training, effectively balancing efficiency and accuracy. This is particularly valuable in deep learning scenarios, where computational resources are intensive, underscoring the importance of downsampling [32]. The downsampling formula is as follows:(4)$$x_{\mathrm{down}}\left[k\right]=\frac{1}{M}\sum_{i=1}^{M}x\left[kM+i\right]$$
where *M* is the downsampling factor (M=10 in this study), x[k] is the original EMG signal, and $x_{\mathrm{down}}\left[k\right]$ is the downsampled signal.

To ensure temporal alignment between the EMG and IMU signals, timestamps were employed to synchronize the two signals. Let the timestamps for the EMG and IMU signals be denoted as $t_{\mathrm{EMG}}$ and $t_{\mathrm{IMU}}$, respectively. The alignment condition is given by(5)$$|t_{\mathrm{EMG}}-t_{\mathrm{IMU}}|<\delta_{t}$$
where δt is the permissible time error ($\delta_{t}=5\;\mathrm{ms}$ in this study). During alignment, signal samples not meeting the synchronization condition were discarded.

Signal segmentation was performed using a fixed-length sliding window approach, dividing the time series data into fixed intervals to generate training and testing samples. Let the sliding window length be *L* and the step size be *S*. The *k*-th window can be expressed as follows:(6)$$X_{k}=\left\{x\left[t_{k}\right],x\left[t_{k}+1\right],\cdots ,x\left[t_{k}+L-1\right]\right\}$$
where $t_{k}=k\cdot S$, *L* is the window length, and *S* is the step size (L=200ms, corresponding to 40 IMU data points; S=50ms in this study). The sliding window segmentation generated continuous time segments, with each segment containing temporally aligned EMG and IMU data.

#### 2.3.2. Feature Extraction from EMG Signals

To construct prediction targets for the model, this study extracted a variety of time-domain and frequency-domain features from the downsampled EMG signals. These features reflect the intensity, frequency, and variation trends of muscle activity, offering a reliable foundation for assessing muscle activation levels. The extracted features include RMS (Root Mean Square), MVA (Mean Absolute Value), ZC (Zero Crossing), WL (Waveform Length), SSC (Slope Sign Change), and MPF (Mean Power Frequency). The specific formulas for these features are given as follows:Root Mean Square (RMS): Reflects the energy intensity of the signal and is calculated as follows:(7)$$\mathrm{RMS}=\sqrt{\frac{1}{N}\sum_{n=1}^{N}x{\left[n\right]}^{2}}$$
where *N* is the number of samples in the signal, and x[n] is the signal amplitude at time *n*.Mean Absolute Value (MAV): Represents the average level of signal amplitude:(8)$$\mathrm{MAV}=\frac{1}{N}\sum_{n=1}^{N}\left|x\left[n\right]\right|.$$Zero Crossing (ZC): Counts the number of times the signal crosses zero within a given threshold ϵ, reflecting the frequency characteristics:(9)$$\mathrm{ZC}=\sum_{n=1}^{N-1}\delta (|x\left[n\right]\cdot x\left[n+1\right]|<\epsilon ),$$
where δ(·) is an indicator function.Waveform Length (WL): Captures the complexity and rate of change of the signal:(10)$$\mathrm{WL}=\sum_{n=1}^{N-1}\left|x\left[n+1\right]-x\left[n\right]\right|.$$Slope Sign Change (SSC): Counts the number of slope changes, reflecting signal fluctuations:(11)$$\mathrm{SSC}=\sum_{n=2}^{N-1}\delta \left((x[n]-x[n-1])\cdot (x[n+1]-x[n])<0\right),$$
where δ(·) equals 1 if the condition is met and 0 otherwise.Mean Power Frequency (MPF): Assesses frequency shifts related to local muscle fatigue:(12)$$\mathrm{MPF}=\frac{\sum_{f=1}^{N}f\cdot P\left(f\right)}{\sum_{f=1}^{N}P\left(f\right)},$$
where P(f) represents the power spectral density at frequency *f*.

These features were calculated within each sliding window, and the resulting values were used to construct the prediction targets for the model. These targets not only quantify the magnitude, frequency, and variation trends of muscle activity but also offer valuable input information for optimizing the model’s performance.

#### 2.3.3. Data Segmentation

After signal processing and feature extraction, the dataset was split into training, testing, and validation sets to ensure the model’s robustness and generalization. During the data splitting process, different movements were split independently, with each repetition in each round treated as an independent sample to ensure the diversity and independence of the dataset.

**Data Splitting Principles** To ensure the independence and representativeness of the data, data splitting was carried out according to the following principles:**Independent Sample Splitting**: Each participant performed three target movements, each with distinct motion characteristics. During the dataset splitting, each individual signal segment $X_{i,j,k,l}$ from a repetition was treated as an independent sample and randomly assigned to the dataset. The structure of the dataset can be expressed as follows:(13)$$D_{i,j}=\overset{K}{\underset{k=1}{⋃}}\overset{L}{\underset{l=1}{⋃}}X_{i,j,k,l},$$
where $X_{i,j,k,l}$ represents the signal segment of the *i*-th participant performing the *j*-th movement in the *k*-th round and the *l*-th repetition.**Movement Group Splitting**: Data from different target movements were independently split into training, validation, and testing sets to ensure the model’s ability to generalize across distinct motion characteristics. Let the complete dataset be D, and the splitting is as follows:(14)$$\left\{\begin{array}{cc} D & =\;\overset{3}{\underset{j=1}{⋃}}D_{j},\\ D_{j} & =\;D_{j,\mathrm{train}}\cup D_{j,\mathrm{val}}\cup D_{j,\mathrm{test}}, \end{array}\right.$$
where $D_{j,\mathrm{train}},D_{j,\mathrm{val}},D_{j,\mathrm{test}}$ represent the training, validation, and testing sets for movement *j*, respectively.**Proportional Splitting**: All independent samples for each movement were allocated as 70% to the training set, 20% to the validation set, and 10% to the testing set:(15)$$\left\{\begin{array}{cc} |D_{j,\mathrm{train}}| & =\;0.70\cdot |D_{j}|,\\ |D_{j,\mathrm{val}}| & =\;0.20\cdot |D_{j}|,\\ |D_{j,\mathrm{test}}| & =\;0.10\cdot |D_{j}|. \end{array}\right.$$**Independence Between Rounds and Movements**: Data from different rounds were kept separate during splitting, ensuring that the samples from each round independently entered different dataset subsets.

During the splitting process, the proportions of different groups (healthy volunteers and participants with shoulder injuries) were preserved to ensure representativeness. Additionally, to prevent data leakage, consecutive data segments within the sliding window were confined to the same subset.

**Data Normalization**: To eliminate amplitude differences across participants, all input data were normalized before modeling. Let the original data be denoted as $x_{i}$, and the normalized data as ${\tilde{x}}_{i}$. The normalization formula is given by:(16)$${\tilde{x}}_{i}=\frac{x_{i}-\mu }{\sigma },$$
where μ is the mean and σ is the standard deviation of the training set. Normalization was based solely on the training set’s statistics to avoid information leakage from the validation and testing sets.

**Final Dataset Organization**: Through the data processing workflow described above, a structured dataset was constructed, containing synchronized IMU and EMG data segments, along with corresponding EMG time-domain feature values. The structure of the dataset obtained after data cleaning is shown in Figure 3. The dataset consists of three subsets: the training set $D_{\mathrm{train}}$, the validation set $D_{\mathrm{val}}$, and the testing set $D_{\mathrm{test}}$. Each subset is balanced in terms of sample distribution, encompassing different movements and rounds from all participants. This dataset organization effectively captures motion characteristic differences, providing high-quality data for model training and validation, as well as a solid foundation for subsequent modeling and prediction.

### 2.4. Public Dataset

In addition to the dataset collected in this study, the SWIFTIES dataset (Subject and Wearables data for Investigation of Fatiguing Tasks in Extension/Flexion of Shoulder/Elbow) [33] was incorporated to enhance the generalizability and applicability of the research and model. SWIFTIES is a high-quality dataset focused on fatigue tasks involving shoulder and elbow flexion/extension movements, containing rich physiological and motion data, making it highly suitable for studies on motion analysis and fatigue detection.

The SWIFTIES dataset includes experimental data from 32 participants (17 males and 15 females), all of whom performed tasks using their dominant hand. The tasks involved shoulder and elbow flexion/extension movements, performed under both static and dynamic conditions at 25% and 45% of Maximum Voluntary Contraction (MVC) force until noticeable fatigue was reached. Each participant completed all combinations of tasks. The experimental design comprehensively covered various motion types, joint areas, and loading conditions, offering diverse scenarios for analyzing muscle activity and movement states.

Following the data processing workflow established in this study, the SWIFTIES dataset underwent similar procedures, such as data alignment, feature extraction, dataset splitting, and standardization. The processed dataset, as shown in Table 2, contains synchronized IMU and EMG data segments, along with EMG time-domain feature values. This dataset provided additional support for the study, effectively enhancing the model’s generalization capabilities and practical value.

## 3. Network Architecture

To comprehensively enhance the feature extraction and sequence modeling capabilities of multi-channel IMU time-series data, this study introduces a novel Transformer-based model architecture, SWCTNet. Designed to address the complexity and diversity of time-series data, SWCTNet integrates the strengths of local feature extraction and global sequence modeling, making it particularly suitable for handling multi-channel, nonlinear, and high-dimensional time-series data. The SWCTNet architecture consists of three key modules: the Sliding Window Convolutional Neural Network Block (SW-CNN Block), the Channel-Time Attention Transformer Block (CTAT Block), and the Downstream Task Block.

The SW-CNN Block utilizes a sliding window mechanism combined with convolutional operations to effectively extract local features from time-series data, capturing short-term patterns and signal variation trends. The CTAT Block, built upon an enhanced multidimensional attention mechanism, focuses on global feature modeling across time steps and channels. The Downstream Task Block is flexibly configured to meet specific application requirements (e.g., classification, regression, or sequence generation), further processing the high-level semantic features output by the CTAT Block to make final predictions or perform generative tasks. The overall architecture of the model is depicted in Figure 4.

While CNN and attention mechanisms have been widely used in tasks such as EMG gesture recognition, the key innovation of SWCTNet lies in its combination of these mechanisms and a sliding window approach that enhances local feature extraction. Specifically, the SW-CNN Block integrates sliding window segmentation with convolution operations to capture fine-grained temporal patterns from multi-channel time-series data, while the CTAT Block introduces a novel channel-time attention mechanism, effectively modeling global dependencies across both time steps and channels. This unique synergy between local feature extraction and global sequence modeling enables SWCTNet to better handle the complexity and diversity of time-series data in applications like multi-channel IMU data analysis.

### 3.1. Sliding Window CNN Block (SW-CNN Block)

The Sliding Window Convolutional Neural Network Block (SW-CNN Block) in SWCTNet is specifically designed to improve the model’s ability to extract local temporal features from multi-channel time-series data, while incorporating inductive biases such as locality and translational invariance into the architecture. These properties are especially crucial for small-scale datasets, as they guide the model to focus on meaningful patterns and minimize the risk of overfitting. By combining a sliding window mechanism with convolutional operations, the SW-CNN Block effectively captures local dependencies, which are critical for both online and offline decoding tasks. Furthermore, the SW-CNN Block enables effective interaction with subsequent Transformer modules by highlighting relevant local temporal features and forwarding them for global dependency modeling. The SW-CNN Block includes the following key steps: sliding window segmentation, one-dimensional convolution (1D-CNN) operations, normalization layers, activation layers, mix pooling operations, and regularization strategies. The architecture of the SW-CNN Block is depicted in Figure 5.

**Sliding Window Partitioning** As a preprocessing step, the sliding window mechanism partitions continuous time-series data into overlapping or non-overlapping local windows. Each window captures a small segment of the signal, allowing for the model to focus on localized temporal features while maintaining the overall structure of the input sequence. The sliding window mechanism also offers a natural form of data augmentation by increasing the number of samples, which is especially beneficial for small datasets. Moreover, this partitioned structure helps isolate transient signal characteristics, enhancing the effectiveness of subsequent convolutional operations.

For input data with dimensions $X\in R^{N_{\mathrm{ch}}\times N_{L}}$, where $N_{\mathrm{ch}}$ represents the number of channels and $N_{L}$ the number of time steps, the sliding window transforms X into $X_{\mathrm{window}}\in R^{L\mathrm{num}\times N_{\mathrm{ch}}\times L_{\mathrm{win}}}$, where $L_{\mathrm{num}}$ is the number of windows and $L_{\mathrm{win}}$ is the window length. The formula for sliding window segmentation is given by(17)$$X_{\mathrm{window}}\left[i\right]=X\left[:,iS:iS+L_{\mathrm{win}}\right],$$
where *S* is the sliding window step size, and *i* is the window index.

**Convolutional Layer** Based on the subsequences generated by the sliding window mechanism, the SW-CNN performs one-dimensional convolution operations on these segmented windows, using convolutional kernels to extract high-level temporal features [34]. The convolution operation slides the kernel along the temporal axis, capturing signal patterns within the window, such as sudden changes, periodic characteristics, or noise variations. The convolution operation is given by(18)$$X_{\mathrm{conv}}\left[k,:,:\right]=\mathrm{ELU}\left(\sum_{c=1}^{C}W_{\mathrm{conv},c}*X_{\mathrm{window}}\left[k,c,:\right]+b_{\mathrm{conv}}\right)$$
where $W_{\mathrm{conv}}$ and *c* represent the learnable kernel weights, $b_{\mathrm{conv}}$ is the bias term, and ELU(·) is the activation function. The use of activation functions such as Exponential Linear Units (ELU) ensures a smooth response to negative inputs, overcoming the limitations of traditional activation functions like ReLU. The incorporation of ELU improves the model’s ability to handle subtle variations in the input data.

**Normalization and Regularization** The SW-CNN module incorporates two regularization strategies, Batch Normalization (BatchNorm) and Window Normalization (WindowNorm), to further improve the model’s training stability and generalization capability. The combination of these techniques not only reduces the impact of non-stationary characteristics in the signal on model performance but also offers targeted optimization for varying input distributions and window-specific features.

BatchNorm is a commonly used normalization strategy in the SW-CNN module, primarily applied to normalize features following convolution operations. The core idea is to normalize each feature channel within a batch to have a mean of 0 and variance of 1, followed by the use of learnable parameters to scale and shift the normalized values [35]. The formula for BatchNrom is given by(19)$${\hat{x}}_{\mathrm{bn}}=\frac{x-\mu_{\mathrm{bn}}}{\sigma_{\mathrm{bn}}},\;y_{\mathrm{bn}}=\gamma_{\mathrm{bn}}\cdot {\hat{x}}_{\mathrm{bn}}+\beta_{\mathrm{bn}},$$
where *x* represents the input features, $\mu_{\mathrm{bn}}$ and $\sigma_{\mathrm{bn}}$ are the mean and standard deviation of the batch, and $\gamma_{\mathrm{bn}}$ and $\beta_{\mathrm{bn}}$ are the learnable scale and shift parameters.

WindowNorm is a normalization strategy specifically designed for sliding windows, which aims to normalize the local features within each window. This approach mitigates the influence of local non-stationary properties on signal feature extraction. Unlike BatchNorm, WindowNorm limits its normalization scope to the features within individual sliding windows. The formula for WindowNorm is given by(20)$${\hat{x}}_{\mathrm{win}}=\frac{x-\mu_{\mathrm{win}}}{\sigma_{\mathrm{win}}},\;y_{\mathrm{win}}=\gamma_{\mathrm{win}}\cdot {\hat{x}}_{\mathrm{win}}+\beta \mathrm{win},$$
where $\mu_{\mathrm{win}}$ and $\sigma_{\mathrm{win}}$ are the mean and standard deviation within the sliding window, and $\gamma_{\mathrm{win}}$ and $\beta_{\mathrm{win}}$ are the learnable scale and shift parameters for the window.

The SW-CNN module incorporates BatchNorm and WindowNorm, enabling it to adapt to global distribution variations while optimizing for local characteristics. This significantly improves the model’s robustness when using small-scale datasets and high-dimensional time-series data.

**MixPooling Layer** To further refine feature maps, the SW-CNN Block incorporates a pooling layer that reduces the spatial dimensions of features, compressing temporal information while retaining the most prominent patterns. A combination of max pooling and average pooling is typically applied, with kernel size and stride configured to effectively downsample the data, reducing the computational cost and preventing overfitting [36].

**Activation and Regularization** Regularization techniques, such as dropout and batch normalization, are essential components of the SW-CNN Block, ensuring robust training and generalization. Dropout is a regularization technique that randomly drops neurons and their connections during training to reduce reliance on specific neurons, thereby preventing overfitting. Mathematically, for the activation values of any layer h, the output after applying dropout becomes:(21)$$h^{\prime }=\frac{h⊙r}{p}$$
where ⊙ denotes element-wise multiplication, r∼Bernoulli(p) is a random vector with each element sampled independently from a Bernoulli distribution with probability *p*, and *p* is the dropout rate. A dropout rate of 0.5 is applied to prevent overfitting, while spatial dropout is employed for highly spatially correlated signals, such as multi-channel time-series data (e.g., IMU signals, EMG signals) [37]. Additionally, a maximum norm constraint is imposed on the weights of each temporal filter, preventing the model from overfitting to noise or irrelevant patterns in the input data.

The SW-CNN Block plays a crucial role in guiding the Transformer components of SWCTNet by providing high-quality local temporal features. These convolutional features not only capture the most relevant patterns within each sliding window but also lay the foundation for learning global dependencies in subsequent Transformer modules. This synergy between the SW-CNN Block and Transformer ensures that both local and global characteristics of time-series data are effectively captured, resulting in improved performance across downstream tasks such as classification, regression, or sequence generation.

### 3.2. Channel-Time Attention Transformer Block (CTAT Block)

The Channel-Time Attention Transformer Block (CTAT Block) in SWCTNet is a key module designed to capture global features and long-range dependencies in multi-channel time-series data. By introducing an enhanced self-attention mechanism, the CTAT Block identifies correlations between time steps and interactions across channels, generating high-level semantic feature representations. The collaboration between the CTAT Block and the SW-CNN Block forms the core of SWCTNet’s feature extraction capabilities. The specific structure of the module is shown in Figure 6.

The input to the CTAT Block comes from the feature map output by the SW-CNN Block, represented as $X\mathrm{cnn}\in R^{k\times H\times L}$, where *k* denotes the number of sliding windows, *H* is the number of channels, and *L* represents the length of the time steps. To meet the input requirements of Transformers, the feature map is reshaped to flatten its dimensions, resulting in $X\mathrm{flat}\in R^{(k\cdot H)\times L}$, which combines features from all channels and windows into a unified input sequence. This transformation provides the foundation for global modeling, enabling the Transformer to focus on both inter-channel and inter-time-step information.

The Channel-Time Attention Mechanism is the core component of the CTAT Block. Built on an improved self-attention mechanism, it consists of two parallel computational paths that focus on capturing dependencies both across channels and over time.

The first is the channel-dominant path, which aims to model the correlations across channels. Specifically, the attention weight matrix $A_{c}$ is computed by taking the inner product of the Query and Key, followed by a Softmax operation, as shown by the following:(22)$$A_{c}=\mathrm{Softmax}\left(\frac{Q_{c}K_{c}^{⊤}}{\sqrt{d_{c}}}\right)$$
where $Q_{c},K_{c}\in R^{L\times d_{c}}$, $d_{c}$ is set to one-fourth of the number of channels to reduce computational complexity.

The second is the time-dominant path, which models long-range temporal dependencies. The attention weight matrix $A_{t}$ is computed by taking the inner product of the Query and Key along the time dimension, followed by a Softmax operation, as expressed by(23)$$A_{t}=\mathrm{Softmax}\left(\frac{Q_{t}K_{t}^{⊤}}{\sqrt{d_{t}}}\right)$$
where $Q_{t},K_{t}\in R^{\left(kH\right)\times d_{t}}$, $d_{t}$ is chosen to be proportional to the original time dimension *L* to ensure sufficient modeling of temporal information.

Finally, the outputs from the two paths are fused using a gating mechanism, which dynamically adjusts the contribution of each attention path based on the input data. The fused attention matrix $A_{\mathrm{attn}}$ is computed as follows:(24)$$A_{\mathrm{attn}}=\lambda A_{c}+\left(1-\lambda \right)A_{t}$$
where the gating coefficient λ is computed via a learnable sigmoid function as follows:(25)$$\lambda =\mathrm{sigmoid}(W_{g}\left[A_{c};A_{t}\right])$$
where $W_{g}$ is a learnable weight matrix. This gating mechanism allows the model to dynamically adjust the contribution of each attention path to the final attention map, thereby optimizing the feature learning process.

To further improve the model’s ability to learn multi-dimensional dependencies, the CTAT Block employs a multi-head attention mechanism, extracting diverse feature subspaces through multiple parallel attention heads. The output of the multi-head attention is combined with the input features using residual connections, followed by layer normalization to stabilize training and enhance optimization efficiency. The fused feature representation is computed as follows:(26)$$X_{\mathrm{fused}}=\mathrm{LayerNorm}\left(X_{\mathrm{attn}}+X_{\mathrm{flat}}\right).$$

To address the absence of positional information in time-series data, the CTAT Block incorporates learnable positional encoding [38]. By combining positional encoding with attention weight matrices, the Transformer captures the sequential characteristics of the time-series data more effectively. The final input sequence is formed by adding feature embeddings (Patch Embeddings) and positional encoding, further improving the model’s perception of sequential data.

By introducing the channel-time attention mechanism, the CTAT Block allows for the more effective global modeling of multi-channel time-series data. Combined with the local features extracted by the SW-CNN Block, the CTAT Block plays a key role in constructing high-quality global feature representations, providing strong support for accurate predictions in downstream tasks.

### 3.3. Downstream Task Block

The Downstream Task Block is the final component of SWCTNet, designed to process features from the CTAT Block and produce the final task-specific outputs. The evaluation of shoulder rehabilitation systems often involves a combination of diverse tasks, including predicting the types of rehabilitation movements, assessing the accuracy of the movement execution, and generating evaluation metrics based on standard movement patterns. Therefore, this block is highly flexible and can be adapted to various tasks, including classification, regression, and sequence generation. With its configurable architecture, the Downstream Task Block enables SWCTNet to be applied in a variety of task scenarios.

The input to the Downstream Task Block consists of global feature representations from the CTAT Block, denoted as $X_{\mathrm{fused}}\in R^{N\times d}$, where *N* is the length of the feature sequence and *d* is the feature dimension. These features integrate the local features extracted by the SW-CNN Block and the global dependencies modeled by the CTAT Block, containing rich spatiotemporal information and inter-channel interactions.

The Downstream Task Block adopts various architectures tailored to specific task objectives. For classification tasks, the block is configured as a Multi-Layer Perceptron (MLP) head to produce class probability distributions. For regression tasks, it uses a linear mapping to predict target values. In sequence generation tasks, a decoder architecture is employed to generate output sequences. The configurations for each task type are outlined as follows:**Classification Tasks** In classification tasks, the Downstream Task Block employs a fully connected layer to map the global features to the class space, followed by a softmax activation function to produce class probabilities. The mathematical expression is given by(27)$$\hat{y}=\mathrm{Softmax}\left(W_{\mathrm{fc}}X_{\mathrm{fused}}+b_{\mathrm{fc}}\right),$$
where $W_{\mathrm{fc}}$ and $b_{\mathrm{fc}}$ are the weights and biases of the fully connected layer, respectively.**Feature Prediction Tasks** For prediction tasks, the module directly outputs predictions through a linear layer:(28)$$\hat{y}=W_{\mathrm{reg}}X_{\mathrm{fused}}+b_{\mathrm{reg}},$$
where $W_{\mathrm{reg}}$ and $b_{\mathrm{reg}}$ are the weights and biases of the regression model. This setup is ideal for tasks such as cross-modal feature mapping and feature prediction.**Sequence Generation Tasks** In sequence generation tasks, the Downstream Task Block employs a decoder architecture to iteratively convert input features into output sequences. The decoder generates the next time-step’s result based on the current input and previous outputs in an autoregressive manner. The expression is given by(29)$${\hat{y}}_{t}=\mathrm{Decoder}\left({\hat{y}}_{t-1},X_{\mathrm{fused}},\theta \right),$$
where $\hat{y}t-1$ is the previous output, Xfused represents the global feature, and θ denotes the decoder parameters.

To prevent overfitting and improve model generalization, the Downstream Task Block uses dropout regularization. The dropout rate is set to 0.5 for classification and regression tasks and 0.7 for sequence generation tasks, depending on task complexity. Additionally, a maximum norm constraint of 0.25 is imposed on the weights in each layer to reduce overfitting to noise or irrelevant features.

The modular design of the Downstream Task Block allows for its seamless adaptation to various task objectives. The classification configuration emphasizes class distinction, making it ideal for recognizing fixed patterns. The regression configuration focuses on continuous value prediction, making it suitable for cross-modal feature mapping or time-series forecasting. The sequence generation configuration supports the generation of complex sequential outputs. By inheriting global features from the CTAT Block and leveraging task-specific designs, the Downstream Task Block enhances predictive capabilities, ensuring an excellent performance across diverse task scenarios.

## 4. Experiments and Results

Based on the model structure and downstream tasks defined above, three groups of experiments were conducted in this study:**Classification Task**: Conducted on the publicly available datasets DB1/DB2/DBA/SWIFITS and the self-constructed dataset, focusing on posture recognition tasks.**Cross-Modal Feature Prediction Task**: Conducted on the SWIFTIES dataset and the self-constructed dataset, aiming to predict EMG features from IMU data.**Cross-Modal Time-Series Generation Task**: Conducted on the SWIFTIES dataset and the self-constructed dataset, focusing on generating EMG time-series data from IMU inputs.

### 4.1. Classification Task

This study proposes a multi-time-series model architecture primarily designed to integrate and analyze multi-channel IMU time-series data for predicting SJ EMG features. To validate the model’s effectiveness and robustness across various multi-time-series tasks, we conducted experiments comparing its performance with baseline models in shoulder motion and posture recognition tasks. For this evaluation, we used three publicly available datasets: NinaPro DB1 (DB1) [39], NinaPro DB2 (DB2) [40], and CapgMyo DB-a (DBA) [41]. Each dataset contains data from different numbers of participants and experimental trials.

Specifically:**DB1 (Ninapro DB1)**: Contains sEMG and kinematic data from 27 participants performing 52 hand movements and a resting position. sEMG signals were recorded using 10 Otto Bock MyoBock electrodes and kinematic data via Cyberglove 2. Movements included finger gestures, isometric/isotonic hand positions, and functional tasks. Each participant completed 10 repetitions, with synchronized sEMG, glove sensor signals, and labeled stimuli.**DB2 (Ninapro DB2)**: Features sEMG, kinematic, inertial, and force data from 40 participants performing 49 hand movements and a resting position. Data were recorded with 12 Delsys Trigno electrodes, Cyberglove 2, accelerometers, and a custom force sensor. The study included finger gestures, grasping motions, and force patterns, with six repetitions per movement.**DBA (CapgMyo DB-a)**: Includes HD-sEMG signals from 18 participants using a 128-channel electrode array (8×16) to capture eight hand gestures. This dataset focuses on high-density sEMG and real-time signal imaging.

A detailed summary of the dataset attributes is provided in Table 3.

The evaluation metrics for the shoulder motion and posture recognition task included class recognition accuracy and overall classification accuracy. Class recognition accuracy is defined as the ratio of correct predictions to the total number of predictions for each category, expressed as follows:(30)$$Accuracy=\frac{\mathrm{Number}\;\mathrm{of}\;\mathrm{correct}\;\mathrm{predictions}}{\mathrm{Total}\;\mathrm{number}\;\mathrm{of}\;\mathrm{predictions}}\times 100\%.$$

The overall mean recognition accuracy (AverageAccuracy) is defined as follows:(31)$$Average\;Accuracy=\frac{1}{N}\sum_{i=1}^{N}Accuracy_{i}$$
where *N* represents the total number of participants. By analyzing these metrics, we evaluated the model’s ability to classify SJ motion postures accurately.

The proposed SWCTNet model showed an exceptional performance in multi-channel time-series classification tasks. To evaluate its effectiveness and robustness in shoulder motion and posture recognition tasks, we conducted experiments using three publicly available datasets (DB1, DB2, and DBA), SWIFITS, and the Personal Dataset. The model’s classification performance was thoroughly compared with several baseline models, as summarized in Table 4.

SWCTNet achieved high accuracy across datasets, with 87.93% on DB1, 88.75% on DB2, and 91.03% on DBA, outperforming several baseline models. On DB1, it effectively balanced local feature extraction via SW-CNN with global dependency modeling through CTAT, outperforming models like CNN-RNN. For DB2, the model’s stability and efficiency in handling complex multi-channel signals surpass state-of-the-art methods, including multi-scale Bi-LSTM and TDCT. On DBA, it excelled in high-dimensional tasks, outperforming methods such as multi-stream CNN and Wavelet Transformer.

For benchmark datasets like SWIFTIES (99.75%), the model exhibited exceptional generalization and adaptability, while on the Personal Dataset (98.00%), it demonstrated superior transferability in personalized scenarios. By integrating local and global feature modeling, the SWCTNet model offers a robust and efficient solution for diverse multi-channel time-series classification tasks, setting new benchmarks for accuracy and robustness.

The SWCTNet excelled in time-series classification, delivering superior accuracy across datasets. By integrating local feature extraction via SW-CNN with global modeling through CTAT, it generated semantically rich representations. This modular design ensures robustness and adaptability, excelling on public datasets, in personalized tasks, and across diverse scenarios. Its performance surpasses state-of-the-art methods, providing a robust foundation for multi-channel analysis and future applications, while delivering efficient solutions for complex time-series tasks with strong generalization capabilities.

### 4.2. Ablation Study

To further investigate the contribution of individual components within the proposed SWCTNet architecture, we conducted an ablation study. The study focused on analyzing the roles of the SW-CNN Block and CTAT Block in overall model performance. The ablation experiments involved systematically removing or modifying specific components to create different variants of the model. The variants included the Transformer Baseline Model (M1), a model with the SW-CNN Block combined with the base Transformer (M2), a model with the CTAT Block combined with the base Transformer (M3), and a model integrating both the SW-CNN Block and the CTAT Block with the base Transformer (M4). Each variant was evaluated on the SJ motion posture recognition task using the same datasets and metrics, allowing for a direct comparison of how each component affected performance. The experimental results for the classification tasks and the ablation study are summarized in Figure 7, offering insights into the contributions of individual components to overall model performance.

The experimental results emphasize the effectiveness of combining the SW-CNN and CTAT Blocks in enhancing classification performance. The baseline model (M1) achieved limited accuracy (e.g., 80.05% on DB1 and 98.31% on SWIFTIES), highlighting the inadequacy of pure global modeling for capturing local features and inter-channel interactions in complex multi-channel time-series data. Incorporating the SW-CNN Block (M2) significantly improved performance by enhancing local feature extraction. For instance, accuracy on DBA increased from 87.39% (M1) to 89.95%, and on SWIFTIES, it improved to 98.92%. The SW-CNN Block effectively captures local dependencies and temporal patterns, demonstrating excellence in fine-grained feature recognition. Integrating the CTAT Block (M3) further enhanced accuracy across datasets by improving global dependency modeling. On DB2, accuracy rose from 80.53% (M1) to 83.91%, and on SWIFTIES, it improved to 98.89%. The CTAT Block leverages channel-time attention to model inter-channel interactions, significantly improving performance on complex signals. The complete model (M4), which combines SW-CNN and CTAT Blocks, achieved the highest accuracy, at 91.03% on DBA and 99.75% on SWIFTIES. This synergy between local and global modeling ensures superior classification accuracy and robustness, enabling the effective handling of diverse and complex time-series tasks.

The ablation study results clearly demonstrate the critical roles of the SW-CNN Block and CTAT Block in local feature extraction and global dependency modeling, respectively. The SW-CNN Block captures local signal features through sliding window convolution, significantly enhancing the model’s sensitivity to fine-grained characteristics. Meanwhile, the CTAT Block strengthens global dependency modeling across channels and time steps via the channel-time attention mechanism. The combination of these modules enables the model to efficiently handle complex multi-channel time-series data, enhancing both classification accuracy and robustness.

### 4.3. Feature Forecast Task

To evaluate the model’s performance in feature prediction, we employed a widely used metric: Root Mean Squared Error (RMSE). RMSE is the square root of the average squared differences between the predicted and true values. A smaller RMSE indicates that the model’s predictions are closer to the observed values. RMSE is calculated as follows:(32)$$\mathrm{RMSE}=\sqrt{\frac{1}{n}\sum_{i=1}^{n}{\left(y_{i}-{\hat{y}}_{i}\right)}^{2}},$$
where $y_{i}$ is the true value, ${\hat{y}}_{i}$ is the predicted value, and *n* is the number of samples.

Based on the sliding window design of the SW-Block, we segmented the aligned EMG data into smaller segments according to the window size. The segmented EMG time-series signals were then preprocessed to compute feature values. These feature values served as inputs to the Decoder, while the encoder–decoder structure of the model was used to predict future feature values. For different pattern categories, the predicted feature values were compared with the true values to compute the RMSE. This evaluation process ensured that the model accurately captured and predicted future EMG feature values across various motion patterns. The radar chart in Figure 8 offers a visual comparison of the model’s performance in the feature prediction task.

In the feature prediction task, aligned EMG data were segmented into multiple time windows according to the SW-CNN Block’s sliding window mechanism. Each window’s signals were preprocessed to compute various feature values, including RMS (Root Mean Square), MAV (Mean Absolute Value), ZC (Zero Crossing Rate), WL (Waveform Length), SSC (Slope Sign Change), and MPF (Mean Power Frequency). These features reflect the signal’s amplitude, frequency, and dynamic variation characteristics.

The model employs an encoder–decoder structure to process input features. The encoder captures both local and global patterns from historical features, while the decoder uses these encoded features to predict future feature values. Discrepancies between predicted and true values are quantified using RMSE.

To provide an intuitive representation of prediction performance, the feature errors were normalized and visualized via radar charts. Figure 8a,b show the feature prediction results for the SWIFTIES dataset and the personal dataset, respectively. In the radar charts, different feature values (e.g., RMS, MAV, ZC) and muscle groups (e.g., Deltoid, Trapezius, Bicep) were normalized to the range [0, 1], where values closer to the circular boundary indicate fewer errors.

On the SWIFTIES dataset, the model exhibited the fewest prediction errors for RMS and MAV features, indicating high accuracy when capturing signal amplitude characteristics. This demonstrates the model’s ability to accurately represent the overall strength and energy of the signal. In contrast, there were more prediction errors for ZC and MPF, indicating that frequency-related features are harder to predict, likely due to transient variations and noise interference. Additionally, prediction performance varied across muscle groups. For example, the Trapezius and Bicep muscles exhibited a better prediction performance than others, likely due to the increased signal stability.

On the personal dataset, the model’s overall prediction performance differed from its performance on the SWIFTIES dataset. Specifically, RMS and MAV prediction errors increased, while ZC and MPF errors slightly decreased. This indicates that amplitude-related features are harder to model on the personal dataset, whereas frequency features demonstrate better adaptability.

Variations in dataset characteristics drive differences in predictions. The personal dataset’s consistent signal acquisition improved its frequency feature predictions (e.g., MPF, ZC) but the model struggled with diverse amplitude features, such as RMS and MAV, compared to the SWIFTIES dataset. Prediction also varied by muscle group; the Bicep and Tricep muscles exhibited better results, while the Latissimus Dorsi showed higher errors in WL and MPF. The model’s sliding window and multi-channel approach effectively captured amplitude, frequency, and dynamic characteristics, ensuring a robust performance across datasets. However, optimizing predictions for specific features and muscle groups under varying conditions remains a key focus for future improvements.

### 4.4. Sequence Generation Task

To demonstrate the effectiveness of our proposed model in multi-time-series sequence generation tasks, we conducted experiments using a personal dataset and the SWIFITS dataset. In these experiments, we fed the actual EMG time-series signals directly into the model as input. The model’s encoder–decoder structure was then used to predict future EMG time-series signals. To evaluate the model’s performance, we employed Dynamic Time Warping (DTW) as an evaluation metric.

DTW is a technique for measuring the similarity between two sequences by finding their optimal alignment. The DTW output represents the distance between two sequences, with a smaller distance indicating higher similarity. The formula for DTW is given by(33)$$\mathrm{DTW}=\min\sum_{i=1}^{n}d\left(y_{i},{\bar{y}}_{j}\right),$$
where $y_{i}$ and ${\bar{y}}_{j}$ denote the two sequences, and *n* represents the length of the sequences. Since the path lengths between different signal pairs may be different, to make the DTW distance comparable, the DTW distance is normalized by dividing the path length, as shown in Equation (Equation 34):(34)$$\mathrm{Normalized}\;\mathrm{DTW}=\frac{\mathrm{DTW}}{n}.$$
where *n* is the length of the sequences.

By employing DTW as evaluation metric, we assessed the model’s ability to generate accurate future EMG time-series signals. These metrics offered insights into the similarity and correlation between the predicted and actual sequences. The visualization in Figure 9 depicts the sequence generation task, presenting partial computational results of time-domain (RMS) and frequency-domain (MPF) features for both the actual and predicted EMG time-series signals.

The figures demonstrate that the model effectively captures overall feature trends, though notable differences exist among different data types. As shown in Figure 9a,c, the model exhibits a superior performance in predicting RMS and MPF features for healthy individuals. The predicted values closely align with the true signals, exhibiting minimal uncertainty and highlighting the model’s strong adaptability to healthy data. For the RMS feature (Figure 9a), the low normalized DTW distance as 0.12 reflects a high level of alignment between predicted and true signals. These results can be attributed to the smoother signal distribution, lower noise levels, and higher correlations in healthy individual data. The model utilizes these characteristics to achieve higher predictive accuracy.

In contrast, the RMS and MPF feature predictions for patient data (Figure 9b,d) exhibit more uncertainty, particularly for the MPF feature (Figure 9d). Patient EMG data often contain more transient changes and irregular fluctuations, resulting in higher DTW distances. For instance, in RMS feature prediction (Figure 9b), while the model captures the overall trend, notable deviations exist in local variations, such as signal peaks. The MPF feature prediction (Figure 9d) demonstrates an even greater error range, underscoring the challenges the model faces in modeling frequency-related features. This phenomenon likely arises from the more complex signal characteristics in patient data, including non-stationary muscle activity, abnormal waveforms, and higher noise levels. Additionally, frequency features in patient data are significantly affected by pathological conditions, complicating the model’s ability to capture consistent patterns and thereby reducing its predictive performance.

The experimental results show that the proposed SWCTNet model performs well in time-series prediction tasks on the personal dataset, but its adaptability differs between healthy and patient data. The predictions for healthy data are more accurate, reflecting the model’s strong ability to handle stationary signals. However, the larger RMS and MPF errors for patient data suggest that complex fluctuations pose greater challenges to the model’s robustness. Future optimization efforts should focus on feature modeling and dynamic prediction for patient data to enhance the model’s adaptability and generalization across diverse tasks.

### 4.5. Discussion

The proposed SWCTNet model demonstrates an exceptional performance in predicting EMG signals from IMU data by effectively capturing both local and global features through its innovative integration of the SW-CNN and CTAT modules. The SW-CNN module extracts local temporal features via sliding window convolution, while the CTAT module employs channel-time attention mechanisms to model the global dependencies across channels and time steps. This synergy enables the robust analysis of multi-channel time-series data, yielding high accuracy for amplitude-related features (e.g., RMS, MAV), as validated by radar charts and time-series generation results that show low errors and strong correlations with actual EMG signals.

A notable strength of SWCTNet is its ability to seamlessly integrate local and global modeling, consistently outperforming baseline models in classification and prediction tasks. Its modular design also allows for it to be adapted to personalized applications across diverse multi-channel datasets. However, while the model performs well with healthy datasets—characterized by stable muscle activation patterns and strong IMU-to-EMG correlations—it encounters challenges with patient data. The increased variability and noise in patient signals lead to higher prediction errors for frequency-related features (e.g., MPF and SSC), highlighting the need for advanced denoising techniques and enhanced frequency-domain modeling to address signal non-stationarity. Moreover, further optimization of computational efficiency and lightweight implementation are necessary to allow for real-time applications on portable devices.

Future research directions include integrating multi-modal data, enhancing frequency-domain modeling through methods such as wavelet transforms or FFT, developing task-specific submodels for personalized predictions, and optimizing hardware adaptation for wearable devices. These enhancements will further solidify the robustness and adaptability of SWCTNet for multi-channel time-series analysis.

### 4.6. Conclusions

This study introduces a novel model architecture based on SW-CNN and a CTAT-Transformer that successfully predicts EMG signal features from multi-channel IMU data. The proposed framework offers a new approach for intelligent evaluation and the personalized rehabilitation of shoulder motion systems. Notably, the findings demonstrate that muscle activation patterns can be accurately predicted using only portable, cost-effective, and easy-to-configure IMU wearable devices—eliminating the need for complex EMG measurement setups. This innovation substantially lowers the barriers for rehabilitation monitoring, providing a convenient and practical solution for home-based rehabilitation. With further improvements in patient data adaptation and multi-modal data integration, the method holds great promise for widespread application in intelligent rehabilitation, motion monitoring, and medical signal processing.

## Figures and Tables

**Figure 1 sensors-25-01275-f001:**
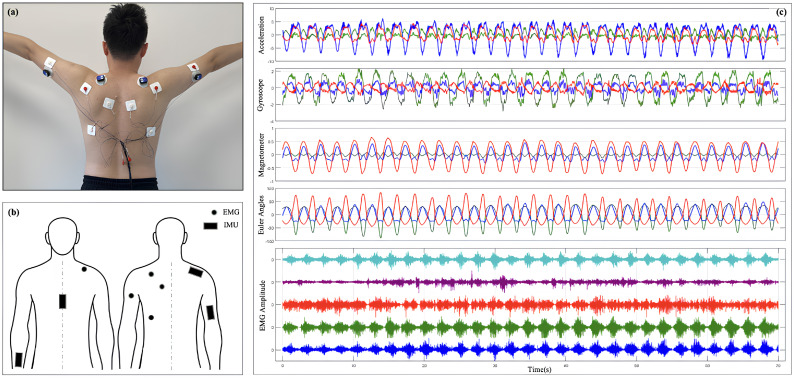
Experimental setup for IMU and sEMG data acquisition: (**a**) placement of the IMU sensor; (**b**) data acquisition checkpoints for IMU and sEMG sensors; (**c**) a subset of raw signals captured by the system.

**Figure 2 sensors-25-01275-f002:**
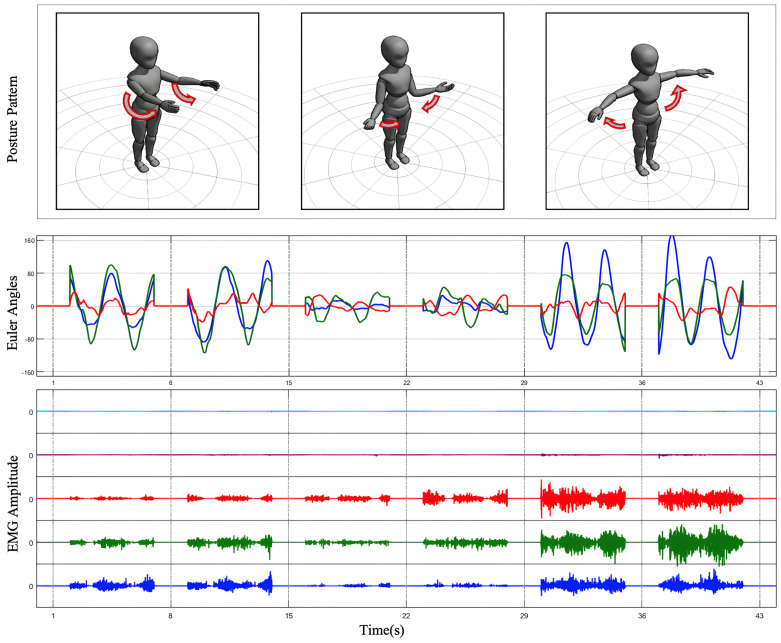
Data process result. The Euler angles obtained from IMU transformation and the processed multi-channel EMG signals are plotted for three preset SJ movements.

**Figure 3 sensors-25-01275-f003:**
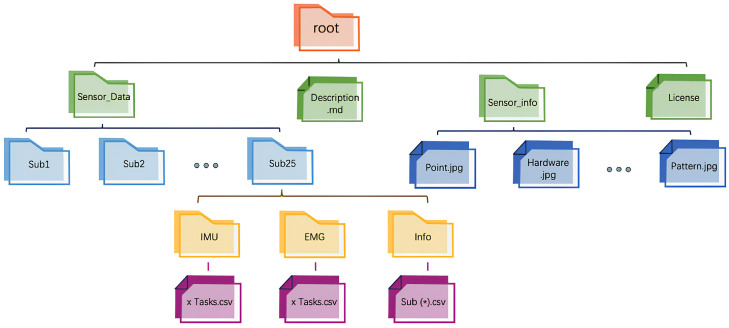
Dataset organization structure.

**Figure 4 sensors-25-01275-f004:**
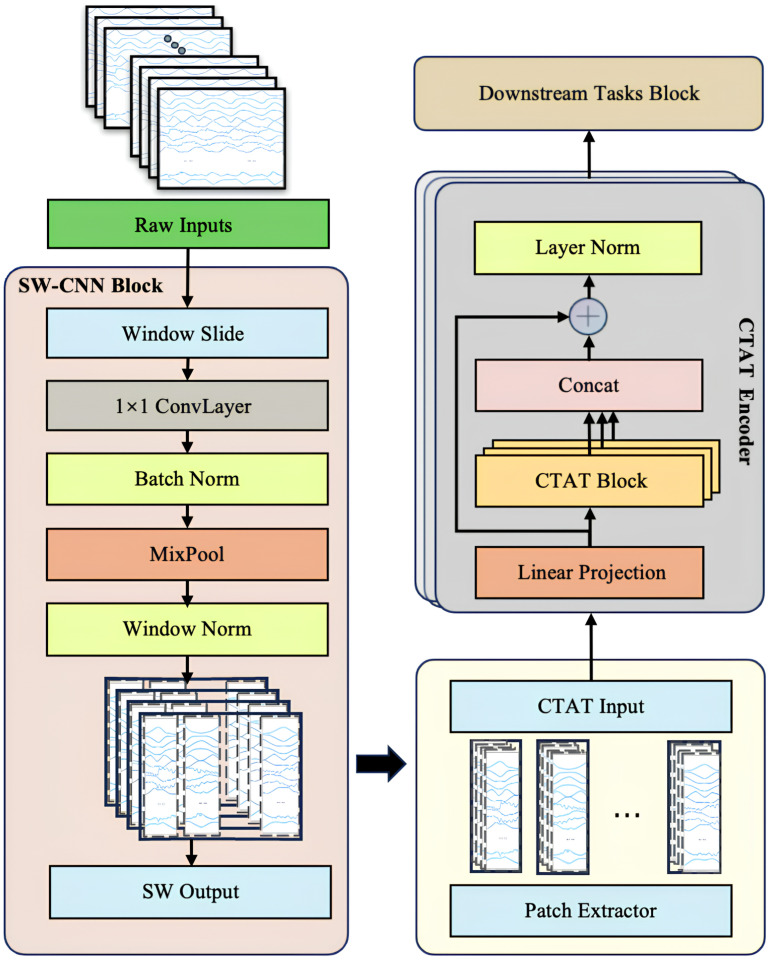
**SWCTNet model architecture.** The model consists of the SW-CNN Block, CTAT Block, and Downstream Task Block.

**Figure 5 sensors-25-01275-f005:**
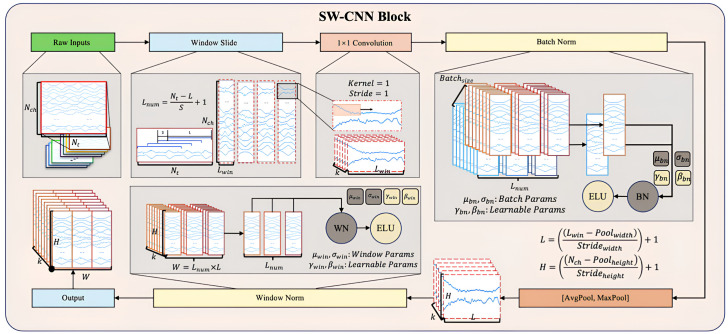
Structure of the SW-CNN Block.

**Figure 6 sensors-25-01275-f006:**
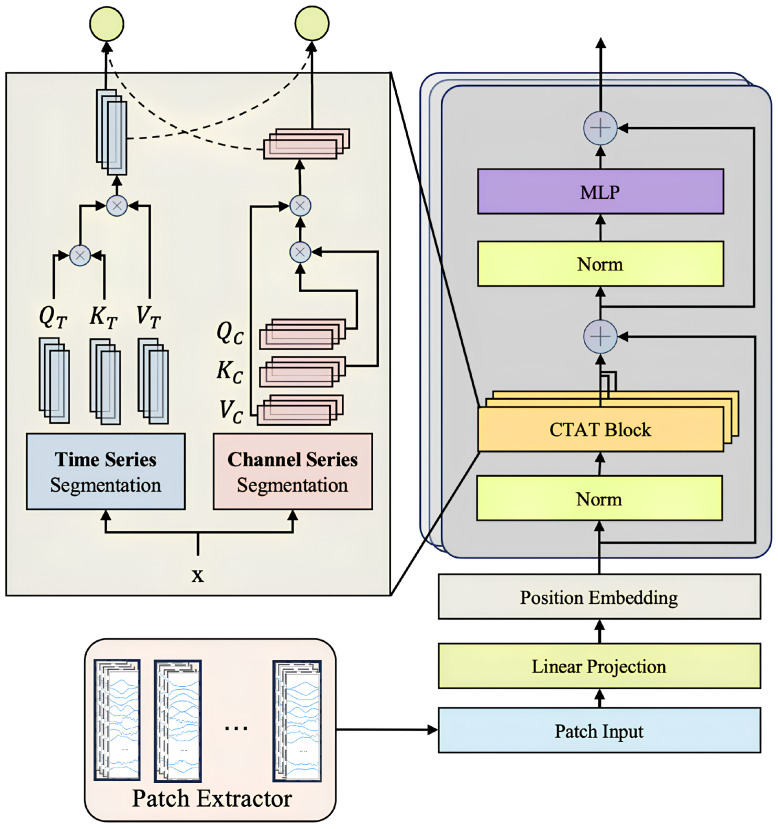
Structure of the CTAT Block.

**Figure 7 sensors-25-01275-f007:**
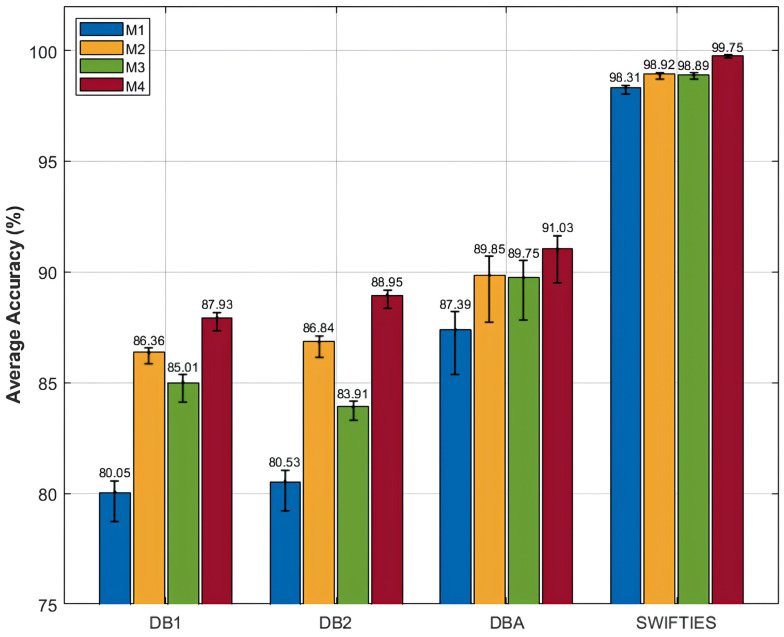
Results of the ablation study conducted on four public datasets.

**Figure 8 sensors-25-01275-f008:**
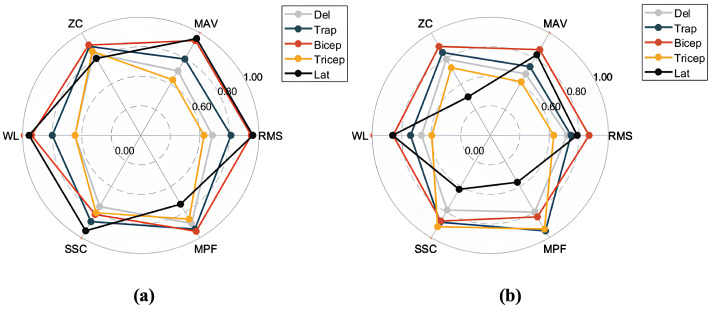
Radar chart comparing the performance of different models on the feature prediction task, normalized to the range [0, 1]: (**a**) results for SWIFTIES dataset; (**b**) results for personal dataset.

**Figure 9 sensors-25-01275-f009:**
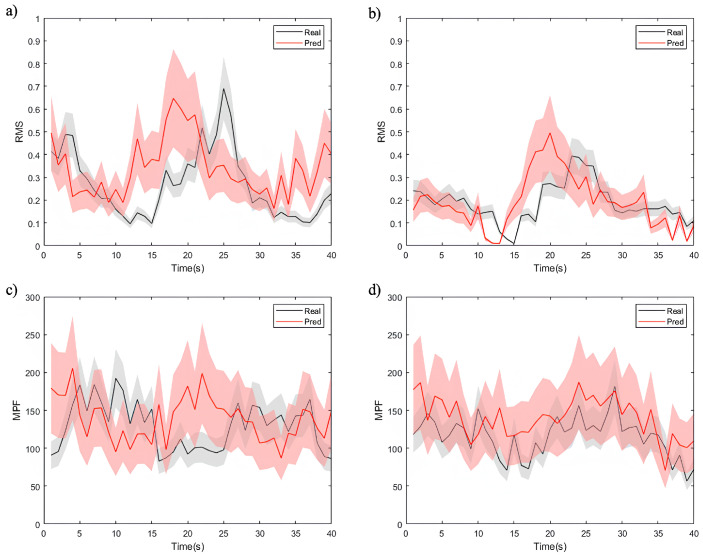
Visualization of the sequence generation task, showing the actual and predicted EMG feature time-series signals, where (**a**,**b**) are the results of the RMS feature, (**c**,**d**) are the results of the MPF feature, (**a**,**c**) are the results for healthy individuals, and (**b**,**d**) are the results for patients.

**Table 1 sensors-25-01275-t001:** Participant information statistics.

Group (n)	Age (Years)	Gender (M/F)	Others
Healthy (10)	25.1 ± 2.9	7/3	-
Injured (15)	25.5 ± 3.2	7/8	fractures mild rotator cuff tears subacromial impingement syndrome

**Table 2 sensors-25-01275-t002:** Processed SWIFTIES dataset with all task combinations and participant information.

Participant	Personal Info	Task Combination	IMU Data (Aligned)	EMG Data (Aligned)	EMG Features (Extracted)
1	Age: 25 Sex: Male Height: 175 cm Weight: 70 kg	Dy/Sh/25%	$X_{1,\mathrm{DySh}25}^{\mathrm{IMU}}$	$X_{1,\mathrm{DySh}25}^{\mathrm{EMG}}$	$F_{1,\mathrm{DySh}25}^{\mathrm{EMG}}$
		Dy/Sh/45%	$X_{1,\mathrm{DySh}45}^{\mathrm{IMU}}$	$X_{1,\mathrm{DySh}45}^{\mathrm{EMG}}$	$F_{1,\mathrm{DySh}45}^{\mathrm{EMG}}$
		Dy/El/25%	$X_{1,\mathrm{DyEl}25}^{\mathrm{IMU}}$	$X_{1,\mathrm{DyEl}25}^{\mathrm{EMG}}$	$F_{1,\mathrm{DyEl}25}^{\mathrm{EMG}}$
		Dy/El/45%	$X_{1,\mathrm{DyEl}45}^{\mathrm{IMU}}$	$X_{1,\mathrm{DyEl}45}^{\mathrm{EMG}}$	$F_{1,\mathrm{DyEl}45}^{\mathrm{EMG}}$
…	…	…	…	…	…
32	Age: 30 Sex: Female Height: 165 cm Weight: 60 kg	Dy/Sh/25%	$X_{32,\mathrm{DySh}25}^{\mathrm{IMU}}$	$X_{32,\mathrm{DySh}25}^{\mathrm{EMG}}$	$F_{32,\mathrm{DySh}25}^{\mathrm{EMG}}$
		Dy/Sh/45%	$X_{32,\mathrm{DySh}45}^{\mathrm{IMU}}$	$X_{32,\mathrm{DySh}45}^{\mathrm{EMG}}$	$F_{32,\mathrm{DySh}45}^{\mathrm{EMG}}$
		Dy/El/25%	$X_{32,\mathrm{DyEl}25}^{\mathrm{IMU}}$	$X_{32,\mathrm{DyEl}25}^{\mathrm{EMG}}$	$F_{32,\mathrm{DyEl}25}^{\mathrm{EMG}}$
		Dy/El/45%	$X_{32,\mathrm{DyEl}45}^{\mathrm{IMU}}$	$X_{32,\mathrm{DyEl}45}^{\mathrm{EMG}}$	$F_{32,\mathrm{DyEl}45}^{\mathrm{EMG}}$

St = Static; Dy = Dynamic; Sh = Shoulder; El = Elbow; 25% = 25% MVC; 45% = 45% MVC.

**Table 3 sensors-25-01275-t003:** Summary of public dataset information. used.

Dataset		Sensor	Sampling Rate (Hz)	Channels	Actions (Repetitions)
NinaPro	DB-1	Otto Bock MyoBock	2000	10	52 hand movements (10 repetitions)
	DB-2	Delsys Trigno	2000	12	49 hand movements (10 repetitions)
CapgMyo	DB-A	HD-EMG (8×16)	1000	128	8 finger movements (10 single trials)

**Table 4 sensors-25-01275-t004:** Accuracy (%) comparison among different models.

Dataset	Reference	Year	Classifier	Accuracy (%)
DB1	Geng [42]	2016	ConvNet	78.90
	Tsagkas [43]	2019	CNN	71.85
	Hu [44]	2019	CNN-RNN	87.00
	Tsinganos [45]	2021	CNN	78.75
	Wang [46]	2022	CNN+GRU	86.00
	Moslhi [47]	2024	Single transformer (FFT)	85.30
	**Ours**	-	**SWCTNet**	**87.93**
DB2	Geng [42]	2016	ConvNet	76.10
	Hu [44]	2018	CNN-RNN	82.20
	Wei [48]	2019	Multi-stream CNN	85.80
	Said [49]	2021	TL + MLP-TL + CNN	67.00
	Wei [50]	2021	Multi-view CNN	83.70
	Kim [51]	2021	CNN-LSTM	83.91
	Shen [52]	2022	CViT	80.02
	Wang [53]	2024	Multi-scale feature+Bi-LSTM	86.66
	Wang [54]	2024	TDCT	87.39
	**Ours**	-	**SWCTNet**	**88.75**
DBA	Geng [42]	2016	ConvNet	89.30
	Wei [48]	2019	Multi-stream CNN	89.50
	Chen [55]	2021	CNN + LSTM + TL	84.77
	Shen [52]	2022	CViT	76.83
	Wang [56]	2023	Self-learning	76.31
	Moslhi [47]	2024	Single transformer (Wavelet)	77.90
	Wang [54]	2024	Position weight	83.00
	**Ours**	-	**SWCTNet**	**91.03**
SWIFITS	**Ours**	-	**SWCTNet**	**99.75**
Personal Dataset	**Ours**	-	**SWCTNet**	**98.00**

## Data Availability

The data presented in this study are available on request from the corresponding author.

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
