# Peer review of "Sliding-Window CNN + Channel-Time Attention Transformer Network Trained with Inertial Measurement Units and Surface Electromyography Data for the Prediction of Muscle Activation and Motion Dynamics Leveraging IMU-Only Wearables for Home-Based Shoulder Rehabilitation"

_sensors, 2025, doi:10.3390/s25041275_

Round 1
Reviewer 1 Report
Comments and Suggestions for Authors
This paper proposes a neural network model architecture called SWCTNet based on Sliding Window Convolutional Neural Network (SW-CNN) and Channel-Time Attention Transformer (CTAT) to address the limitation of relying on spatial motion data when using Inertial Measurement Units (IMUs) for shoulder rehabilitation. The paper is well-structured overall, but there are several issues that need to be addressed before considering publication.
1. The major innovation of this paper is the proposed SWCTNet model (Sliding Window CNN + Channel-Time Attention Transformer). However, the combination of CNN and attention mechanisms has been widely used in EMG gesture recognition. The authors should focus on clearly clarifying he specific improvements of their model.
2. The study aims to predict EMG features from IMU data. How is the synchronization and alignment of these two signals ensured during the process of handling and integrating the signals for direct use?
3. The paper spends a lot of space on model construction and parameter selection but fails to provide a detailed introduction to the key component, the Channel-Time Attention Transformer module.
4. The research aims to predict EMG features from IMU data for shoulder rehabilitation, but Section 4.1 uses hand movement data to evaluate the model's performance, which seems inconsistent with the research focus. Please provide a clear explanation for this choice.
5. There is overlap between the conclusion and discussion sections, which should be revised for clarity.
6. The introduction in Chapter 1 is overly verbose and should be condensed.
7. The paper should include more structural diagrams and flowcharts to more clearly express the proposed structures and processes.
Author Response
Comments 1: The major innovation of this paper is the proposed SWCTNet model (Sliding Window CNN + Channel-Time Attention Transformer). However, the combination of CNN and attention mechanisms has been widely used in EMG gesture recognition. The authors should focus on clearly clarifying he specific improvements of their model.
Response 1: Thank you for pointing this out. The clarification has been provided on page 12 (lines 380-388).
Comments 2: The study aims to predict EMG features from IMU data. How is the synchronization and alignment of these two signals ensured during the process of handling and integrating the signals for direct use?
Response 2: Data alignment is indeed an important issue in multimodal sensor data collection, and we have provided a brief explanation in the ‘Data Alignment’ section on page 7. Furthermore, we primarily rely on hardware timestamps for alignment and perform additional data dimension alignment through downsampling. The hardware design and methods are already well-established, and this paper mainly focuses on the subsequent algorithm design, so we do not elaborate on the hardware part in detail.
Comments 3: The paper spends a lot of space on model construction and parameter selection but fails to provide a detailed introduction to the key component, the Channel-Time Attention Transformer module.
Response 3: Upon further review, we agree that the explanation of the CTAT module was indeed insufficient. Therefore, we have revised and expanded this section, specifically in the ‘CTAT Block’ section on page 16 (lines 487-538).
Comments 4: The research aims to predict EMG features from IMU data for shoulder rehabilitation, but Section 4.1 uses hand movement data to evaluate the model's performance, which seems inconsistent with the research focus. Please provide a clear explanation for this choice.
Response 4: From the perspective of the experimental categories, this section is not directly related to the core research. However, the essence of both tasks lies in the design and processing of multi-channel time-series data. We present the results of this section to demonstrate the model’s effectiveness in classification tasks within this domain. Additionally, training on multi-channel time-series tasks provides a pre-trained model for this study, facilitating the subsequent overall training process.
Comments 5: There is overlap between the conclusion and discussion sections, which should be revised for clarity.
Response 5: Thank you for your comment. We have revised the Discussion and Conclusion sections to better differentiate them from each other, from page 25 to 26 (lines 797-834).
Comments 6: The introduction in Chapter 1 is overly verbose and should be condensed.
Response 6: We have made some reductions to the Introduction section, from page 1 to 3 (lines 21-117).
Comments 7: The paper should include more structural diagrams and flowcharts to more clearly express the proposed structures and processes.
Reviewer 2 Report
Comments and Suggestions for Authors
Dear Authors,
Thank you for the opportunity to review your manuscript. I find your work very interesting, especially in the context of ROM and muscles activity. To improve the clarity of your work, I suggest the following:
-
Title – Consider changing the title of your work. The current title is confusing and not informative.
-
Abstract – The abstract contains dense technical jargon and would benefit from more clarity and conciseness.
Some phrases in the abstract are awkward, such as:
- "Additionally, SWCTNet exhibits remarkable precision and stability in generative tasks, that the normalize DTW distance get 0.12 for the normal group and 0.25 for the patient group on self-collected dataset."
- Did you mean: "The normalized DTW distance was 0.12 for the normal group and 0.25 for the patient group on the self-collected dataset."
-
Introduction – Some sentences are too long and complex, making comprehension difficult.
- The phrase "shoulder joint" appears 15 times in the text—consider using the abbreviation SJ to improve readability.
- The introduction contains excessive background information on shoulder anatomy and injury prevalence. Some of this could be shortened or even removed to improve clarity.
-
Materials and Methods –
- Before presenting the experimental setup image, introduce the sensors to ensure a logical flow.
- The placement of Figures 1 and 2 needs to be reconsidered to ensure they are referenced in the text before they appear.
- The methodology, while detailed, lacks clarity in the experimental setup description.
- Figure descriptions are insufficient (e.g., Figure 2). Additionally, many figures are too small to read and analyze.
-
Manuscript Formatting – Please ensure that the manuscript adheres to the recommended Research Manuscript Sections outlined by the MDPI Sensors Journal:
MDPI Sensors Journal Guidelines. -
Spelling Error – "2nd Affliated Hospital" should be corrected to "2nd Affiliated Hospital."
I look forward to the revised version of your manuscript.
Best regards
Comments on the Quality of English Language
There are several issues here and there. I suggested some changes in my review.
Author Response
Comments 1: Title – Consider changing the title of your work. The current title is confusing and not informative.
Response 1:The title will be further revised after confirmation with the editor.
Comments 2: Abstract – The abstract contains dense technical jargon and would benefit from more clarity and conciseness.Some phrases in the abstract are awkward, such as:
"Additionally, SWCTNet exhibits remarkable precision and stability in generative tasks, that the normalize DTW distance get 0.12 for the normal group and 0.25 for the patient group on self-collected dataset."Did you mean: "The normalized DTW distance was 0.12 for the normal group and 0.25 for the patient group on the self-collected dataset."
Response 2: Thank you for your comment. I have made revisions to this section (line 12).
Comments 3:Introduction – Some sentences are too long and complex, making comprehension difficult.
The phrase "shoulder joint" appears 15 times in the text—consider using the abbreviation SJ to improve readability.
The introduction contains excessive background information on shoulder anatomy and injury prevalence. Some of this could be shortened or even removed to improve clarity.
Response 3:Thank you for your comment. Indeed, this section was somewhat verbose. We have made some reductions to this part and also improved the use of abbreviations, from page 1 to 3 (lines 21-117).
Comments 4:Materials and Methods -
Before presenting the experimental setup image, introduce the sensors to ensure a logical flow.
The placement of Figures 1 and 2 needs to be reconsidered to ensure they are referenced in the text before they appear.
The methodology, while detailed, lacks clarity in the experimental setup description.
Response 4: We have further confirmed the image presentation and content sequence. Additionally, we have supplemented the images of the sensors.
Comments 5: Figure descriptions are insufficient (e.g., Figure 2). Additionally, many figures are too small to read and analyze.
Response 5: Thank you for your comment. We have improved the caption accordingly in page 6 (before line 200).
Comments 6: Manuscript Formatting – Please ensure that the manuscript adheres to the recommended Research Manuscript Sections outlined by the MDPI Sensors
Response 6: Thank you for your reminder. We have strictly followed the recommended format since the beginning of drafting the manuscript. We will conduct a further check during this round of submission.
Comments 7: Spelling Error – "2nd Affliated Hospital" should be corrected to "2nd Affiliated Hospital."
Response 7: Thank you for your reminder. This section indeed had an incorrect spelling due to oversight, and we have corrected it (page 1).
Reviewer 3 Report
Comments and Suggestions for Authors
The article presents a substantial amount of work with good innovation, but it is somewhat lengthy. The description of basic concepts could be reduced. Additionally, there are a few issues that require attention.
1. The authors have opted for a 1×1 convolution, stating that it is used to capture local features. However, it seems challenging to achieve this with a 1×1 convolution alone. Have the authors considered experimenting with convolutions of other sizes?
2. The setup for model training requires further elaboration. The authors propose a highly versatile backbone, and the model's pretraining plays a significant role in its performance. Was the model pretrained? Additionally, was the model in the comparative experiments adequately trained? Some comparative experiments on a self-built dataset would also be valuable.
3. The authors mention that certain works perform well but are computationally expensive. Could the authors provide information on the number of parameters in their proposed model? How does the number of parameters compare to other models in existing research? Including this information would help clarify whether the observed performance improvement is due to an increase in parameter count.
4. There is a typographical error in the caption of Figure 4, where "CCTAT Block" is mentioned.
Author Response
Comments 1: The authors have opted for a 1×1 convolution, stating that it is used to capture local features. However, it seems challenging to achieve this with a 1×1 convolution alone. Have the authors considered experimenting with convolutions of other sizes?
Response 1: For multi-channel time-series tasks, the choice and implementation of convolutional kernels is complex and challenging. Initially, we implemented kernels of size 3 or 5, but the results were not satisfactory. We believe this may be due to some redundancy introduced by the sliding window processing. In fact, we are still working on optimizing the structure and parameters of the convolutional part.
Comments 2: The setup for model training requires further elaboration. The authors propose a highly versatile backbone, and the model's pretraining plays a significant role in its performance. Was the model pretrained? Additionally, was the model in the comparative experiments adequately trained? Some comparative experiments on a self-built dataset would also be valuable.
Response 2: The initial model was unable to retrieve similar structural designs or solutions in this field. Therefore, we began training the model from scratch for the initial classification task. The subsequent pre-training part mainly involved transitioning between different tasks (from classification to prediction, and finally to generative tasks). As for whether sufficient training has been conducted, the answer might be negative based on the current results. However, the model already demonstrates adequate performance after light fine-tuning. In fact, we are still further configuring and training the model, having deepened the structure of the main model to meet the further needs of rehabilitation applications.
Additionally, in the comparison experiments with our self-constructed dataset, we presented two intuitive comparison results, which cover both the time-domain and frequency-domain aspects.
Comments 3: The authors mention that certain works perform well but are computationally expensive. Could the authors provide information on the number of parameters in their proposed model? How does the number of parameters compare to other models in existing research? Including this information would help clarify whether the observed performance improvement is due to an increase in parameter count.
Response 3: Based on the previous reply, we are still revising it. At the same time, the main architecture of the model has already been further trained with a deeper structure. Following the main workflow described in the paper, this model process is able to achieve good results.
Comments 4: There is a typographical error in the caption of Figure 4, where "CCTAT Block" is mentioned.
Round 2
Reviewer 2 Report
Comments and Suggestions for Authors
No comments